# Field-Testing a Proteomics-Derived Machine-Learning Model for Predicting Coral Bleaching Susceptibility

Anderson B. Mayfield [1,2,*] and Chiahsin Lin [3,4]

1   Coral Reef Diagnostics, Miami, FL 33129, USA
2   International Society for Reef Studies, Tavernier, FL 33070, USA
3   National Museum of Marine Biology and Aquarium, Checheng 944, Pingtung, Taiwan
4   Institute of Marine Biology, National Dong-Hwa University, Checheng 944, Pingtung, Taiwan
*   Correspondence: anderson@coralreefdiagnostics.com; Tel.: +1-337-501-1976

**Featured Application: An artificial intelligence trained with protein concentration data was able to accurately predict the bleaching susceptibility of massive corals of the Upper Florida Keys (USA).**

**Abstract:** Given the widespread decline of coral reefs, temperature-focused models have been generated to predict when and where bleaching events may occur (e.g., Coral Reef Watch). Although such algorithms are adept at forecasting the onset of bleaching in many areas, they suffer from poor predictive capacity in regions featuring corals that have adapted or acclimatized to life in marginal environments, such as reefs of the Florida Keys (USA). In these locales, it may instead be preferred to use physiological data from the corals themselves to make predictions about stress tolerance. Herein proteomic data from both laboratory and field samples were used to train neural networks and other machine-learning models to predict coral bleaching susceptibility in situ, and the models' accuracies were field-tested with massive corals (*Orbicella faveolata*) sampled across a 2019 bleaching event. The resulting artificial intelligence was capable of accurately predicting whether or not a coral would bleach in response to high temperatures based on its protein signatures alone, meaning that this approach could consequently be of potential use in delineating *O. faveolata* climate resilience.

**Keywords:** artificial intelligence; coral reefs; dinoflagellates; global climate change; machine-learning; molecular biotechnology; proteomics; temperature

## 1. Introduction

The increasing pace of seawater temperature rise has led to the search for environmentally hardy corals that could be used in restoration initiatives [1]. However, the current means by which coral climate resilience is assessed is retroactive: via documenting the degree of bleaching in stress-prone individuals during surveys or lab experiments [2,3]. By the time all reefs are surveyed (or all species experimentally tested in the lab), many others will have perished [4]. Ideally, the health of a coral could be known *prior to* late-stage manifestations of sickness, but at present no such proactive means for predicting coral fate exists despite a large body of literature on the eco-physiology of corals [5,6]; in most cases, the implicit goal of these studies is to improve predictions of how coral reefs will change in the coming decades, though in no cases to date have molecular eco-physiological data been used to develop analytical tools for forecasting coral persistence.

This could stem from the risk of using lab data to make inferences about animal behavior in situ. Even reciprocal transplants and field-based environmental challenge studies inherently create conditions to which corals would never be exposed (e.g., coring with a drill). Nevertheless, insight into the molecular biology of reef corals gained from both in situ and ex situ experiments could perhaps be useful in predictive model building. As an example, if a particular gene is only expressed by bleaching-susceptible corals in the weeks–months prior to onset of bleaching, and never by those that resist this phenomenon,

then this mRNA could serve as a biomarker for environmental stress susceptibility that would be useful in formulating predictions [7,8]. In a series of works on the well-studied, framework-building, relatively resilient Caribbean reef coral *Orbicella faveolata* [9,10], high-temperature tolerance was found to vary across shelves (inshore vs. offshore), as well as among genotypes. In addition to transcriptome profiling [9], two proteomics approaches, shotgun sequencing [10] and isobaric tags for relative and absolute protein quantification (iTRAQ [11]), were utilized to understand the cellular basis underlying this heterogeneity in thermotolerance observed in situ; proteins involved in lipid trafficking, immunity, and other pathways were found to be important in determining the degree to which distinct genotypes resisted high-temperature-induced bleaching [12,13].

Despite the insight gained from this multi-'Omic approach, the machine-learning (ML) models trained from these datasets were only validated with lab data held back from the artificial intelligence (AI). Given that the experiment featured a mix of coral genotypes from different environments that displayed a range of high-temperature tolerances, we hypothesized that we could use the proteomic data to develop models capable of predicting the bleaching susceptibility of corals in situ. To field-test these models, we sampled replicate *O. faveolata* genotypes from each of two inshore and two offshore reef sites of the Upper Florida Keys before, during, and after an anomalously high-temperature event in the summer of 2019 (Figure 1), analyzed their proteomes, input the proteomic data into both the ML models of a prior work [11] and those newly constructed herein, and hypothesized that we could diagnose coral bleaching susceptibility at an accuracy > 80%.

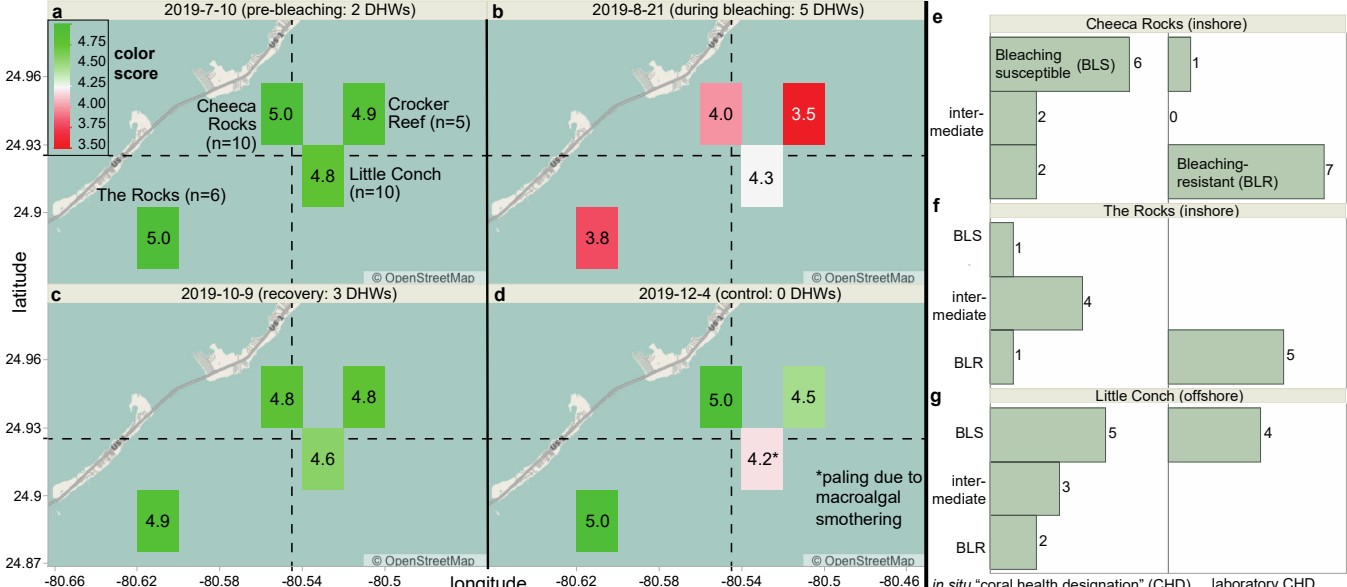

**Figure 1.** Map of study sites (**a**–**d**) and breakdown of field-tracked and lab coral phenotypes (**e**–**g**). Field sample sizes and the color score legend are both in a. DHW = degree-heating weeks. Panels (**e**–**g**) present the breakdown of samples from three of the four study reefs (excluding the test site, Crocker Reef) with respect to their bleaching susceptibility, and the values adjacent to the columns signify number of colonies (rather than genotypes) since, in certain cases, colonies of the same genotype demonstrated different thermotolerances. The colony health designations (CHD) of lab corals [11] from these same reefs have also been shown. Note that the five field-tracked colonies from Crocker Reef, which did not feature in the experiments, were characterized by the following CHD: bleaching-resistant (n = 1), intermediate (n = 2), unknown (n = 1; unable to locate colony at all survey times), and diseased (n = 1; succumbed to stony coral tissue loss disease & was excluded from analysis).



## 2. Materials and Methods

### 2.1. Approach and Terminology

We sought to predict the coral response to elevated temperature exposure (Figure 2a) as defined by a response metric called the "colony health designation" (Figure 2b–e); this parameter could be bleaching-susceptible, bleaching-resistant, or intermediate based on colony color score [14] decreases of $\geq 2$, 0, and 1, respectively. The colony health designation is distinct from the "fragment health designation" defined previously [11], which was instead the health status of a biopsy at the time of sampling. A bleaching-susceptible coral (Figure 2c,e) could be characterized by three of the four fragment health designations: (1) healthy control (no color change while at a temperature below the mean monthly maximum [MMM]), (2) sub-lethally stressed (color score decrease of 1 while at a temperature $\geq$ MMM + 1 °C), or (3) actively bleaching (color score change $\geq 2$ at a temperature $\geq$ MMM + 1 °C). A bleaching-resistant coral (Figure 2b,d) could be a heathy control, present a sub-lethally stressed phenotype, or be deemed "high-temperature-acclimating" (no color change while at a temperature $\geq$ MMM + 1 °C; pooled with healthy controls for most analyses); it would never be associated with an actively bleaching phenotype. Although not depicted in Figure 2, an intermediately bleaching-susceptible coral could generate a biopsy of any of the four fragment health designations.

The capacity to delineate coral resilience in situ was grown through the following steps. First, we used partial least squares-discriminant analysis (PLS-DA) to differentiate lab coral biopsies of three fragment health designations (healthy controls & high-temperature-acclimating samples were combined for this analysis). Predicting the bleaching susceptibility of a perceivably healthy coral colony (step 2) was more challenging and involved training an AI with data from "healthy" biopsies from corals that might later respond differently to high temperatures. To this end, an AI was first trained with proteomic data from samples harvested earlier in the temperature challenge study (days 0–5; described below) to make predictions on their bleaching susceptibility (typically manifested between days 10 & 30) based on their proteomic signatures. We next used PLS-DA to differentiate actively bleaching, sub-lethally stressed, and healthy control biopsies collected across a 2019 bleaching event (described below) via their proteomic profiles. These models ensured that the AI could distinguish corals of the various fragment health designations.

For step 4 (the primary goal), we took three approaches to predict the fates of field corals based on their protein biomarker levels. First, data exclusively from the lab study [11] (Table 1) were used to train predictive models ("CHD-lab➔CHD-field"), which were then tested with data from field corals sampled across a bleaching event (Figure 1 and Table 2). Secondly, field proteomic data exclusively were used to train predictive models of colony bleaching susceptibility ("CHD-field➔CHD-field"). Thirdly, we combined both lab and field sample data into an ensemble model for predicting field coral behavior.

### 2.2. The Experiment

The temperature challenge study was described previously [9,10]. Briefly, genotyped *O. faveolata* colonies from three well-characterized Upper Florida Keys long-term monitoring sites [12]—Cheeca Rocks (inshore), The Rocks (inshore), and Little Conch (offshore; Table 1)—were cored with a pneumatic drill to generate ~5-cm-diameter "pucks." The pucks were allowed to recover in situ and then in the lab, and later exposed to either 33 °C for five days or 32 °C for 31 days (vs. controls at the ambient temperature at time of sampling: 30 °C). A detailed treatise of the climatology of the field sites can be found in [13]; briefly, corals of these sites normally begin accruing heat stress above 31.3 °C, rather than the MMM (31 °C) + 1 °C (32 °C); degree-heating week (DHW) calculations were instead made based on this threshold. The proteomes of a subset of samples (Table 1) were analyzed using one of two methodologies (with ~75% of biopsies analyzed by both): shotgun proteomics [10] and/or iTRAQ [11]. Given the low degree of overlap between the two technologies, we focused on iTRAQ herein since it is the more quantitative of the two. Additional details of the laboratory experiment can be found in Supplementary File S1.

**Table 1.** Details of the 20 lab samples. This table bears similarity to Table 1 of [11] except that an additional colony health designation (CHD) column has been included for field samples. Note that the CHD was not always the same ex situ vs. in situ. Asterisks (*) denote genotypes whose field proteomes were assessed (Table 2). Abbreviations: AB = actively bleaching (n = 2). BLS = bleaching-susceptible, BLR = bleaching-resistant, INT = intermediate (field samples only), and FHD = fragment health designation. HC = healthy controls, HTA = high-temperature-acclimating (pooled with HC for most analyses), ID = indeterminant (not enough data to determine), NA = not applicable, and SLS = sub-lethally stressed.

| Sample Name | Reef of Origin | Shelf | Treatment (Temp.-Time) | Genotype | CHD Field | CHD Lab | FHD Lab | Protein (µg) | iTRAQ Tag | iTRAQ Batch |
|---|---|---|---|---|---|---|---|---|---|---|
| Normalizer | mix of all | mix of both | mix of all | mix of all | NA | NA | NA | 22 | 113 | A |
| B5-7 [a] | Cheeca Rocks | inshore | 30-5 | lightyellow | BLS | BLR | HC | 22 | 114 | A |
| C5-1 | Little Conch | offshore | 30-5 | black(e) | BLS | BLS | HC | 22 | 115 | A |
| B5-4 | Cheeca Rocks | inshore | 33-5 | lightyellow | BLS | BLR | SLS | 22 | 116 | A |
| A2-2 | The Rocks | inshore | 30-31 | skyblue * | BLR | BLS | HC | 22 | 117 | A |
| A4-5 | The Rocks | inshore | 32-31 | skyblue * | INT | BLR | HTA | 22 | 118 | A |
| B3-1 | Cheeca Rocks | inshore | 32-31 | black(c) | BLR | BLR | HTA | 22 | 119 | A |
| C5-2 | Little Conch | offshore | 32-31 | black(e) | BLS | BLS | AB | 22 | 121 | A |
| Normalizer | mix of all | mix of both | mix of all | mix of all | NA | | NA | 22 | 113 | B |
| A4-1 | The Rocks | inshore | 30-5 | skyblue * | INT | BLR | HC | 22 | 114 | B |
| C2-2 | Little Conch | offshore | 30-5 | black(b) | BLS | BLS | HC | 22 | 115 | B |
| D5-2 | Cheeca Rocks | inshore | 30-5 | grey60 | BLS | BLR | HC | 22 | 116 | B |
| D6-6 | Cheeca Rocks | inshore | 33-5 | grey60 | BLS | ID | SLS | 22 | 117 | B |
| A4-8 | The Rocks | inshore | 30-31 | skyblue * | INT | BLR | HC | 22 | 118 | B |
| C5-7 | Little Conch | offshore | 30-31 | black(e) | BLS | BLS | HC | 22 | 119 | B |
| D5-3 | Cheeca Rocks | inshore | 32-31 | grey60 | BLS | BLR | HTA | 22 | 121 | B |
| Normalizer | mix of all | mix of both | mix of all | mix of all | NA | | NA | 22 | 113 | C |
| A4-7 | The Rocks | inshore | 33-5 | skyblue * | INT | BLR | HTA | 22 | 114 | C |
| C5-8 | Little Conch | offshore | 33-5 | black(e) | BLS | BLS | AB | 22 | 115 | C |
| D5-5 | Cheeca Rocks | inshore | 33-5 | grey60 | BLS | BLR | HTA | 22 | 116 | C |
| B5-1 | Cheeca Rocks | inshore | 30-31 | lightyellow | BLS | BLR | HC | 22 | 117 | C |
| D4-8 | Cheeca Rocks | inshore | 30-31 | grey60 | BLS | BLR | HC | 22 | 118 | C |
| D5-8 | Cheeca Rocks | inshore | 30-31 | grey60 | BLS | BLR | HC | 22 | 119 | C |
| B5-2 | Cheeca Rocks | inshore | 32-31 | lightyellow | BLS | BLR | SLS | 22 | 121 | C |

[a] Sample was compromised during processing.

**Table 2.** Details of the 36 field samples. Colony health designations (CHD) were bleaching-resistant (BLR; n = 12), bleaching-susceptible (BLS; n = 12), or intermediate (INT; n = 12). For fragment health designation (FHD) abbreviations, see Table 1; HC, HTA, SLS, and AB sample sizes were 22, 4, 5, and 4, respectively. One colony instead succumbed to stony coral tissue loss disease (SCTLD). NA = not applicable.

| Sample Name | Reef of Origin | Shelf | Sampling Date | Genotype | CHD Field | FHD Field | Protein (μg) | iTRAQ Tag |
|---|---|---|---|---|---|---|---|---|
| **Batch A** | | | | | | | | |
| Normalizer-A | mix of all | mix of both | mix of all | mix of all | NA | NA | 58 | 113 |
| Cheeca-3132-7/19 | Cheeca Rocks | inshore | 2019-7 | black(a) | BLR | HC | 70 | 114 |
| Cheeca-3916-7/19 | Cheeca Rocks | inshore | 2019-7 | lightgreen | BLS | HC | 70 | 115 |
| Cheeca-3939-7/19 | Cheeca Rocks | inshore | 2019-7 | darkred | INT | HC | 70 | 116 |
| CR-3165-7/19 | Crocker Reef | offshore | 2019-7 | black(r) | BLR | HC | 70 | 117 |
| CR-3679-7/19 | Crocker Reef | offshore | 2019-7 | not genotyped | INT | HC | 70 | 118 |
| CR-3986-7/19 | Crocker Reef | offshore | 2019-7 | black(aa) | BLS | HC | 70 | 119 |
| **Batch B** | | | | | | | | |
| Normalizer-B | mix of all | mix of both | mix of all | mix of all | NA | NA | 58 | 113 |
| LC-3694-7/19 | Little Conch | offshore | 2019-7 | black(f) | BLS | HC | 70 | 114 |
| LC-3921-7/19 | Little Conch | offshore | 2019-7 | black(i) | BLR | HC | 70 | 115 |
| LC-3989-7/19 | Little Conch | offshore | 2019-7 | black(g) | INT | HC | 70 | 116 |
| Rocks-3105-7/19 | The Rocks | inshore | 2019-7 | skyblue | BLS | HC | 70 | 117 |
| Rocks-3148-7/19 | The Rocks | inshore | 2019-7 | skyblue | INT | HC | 70 | 118 |
| Rocks-3906-7/19 | The Rocks | inshore | 2019-7 | skyblue | BLR | HC | 70 | 119 |
| **Batch C** | | | | | | | | |
| Normalizer-C | mix of all | mix of both | mix of all | mix of all | NA | NA | 58 | 113 |
| Cheeca-3132-8/19 | Cheeca Rocks | inshore | 2019-8 | black(a) | BLR | HTA | 70 | 114 |
| Cheeca-3916-8/19 | Cheeca Rocks | inshore | 2019-8 | lightgreen | BLS | AB | 70 | 115 |
| Cheeca-3939-8/19 | Cheeca Rocks | inshore | 2019-8 | darkred | INT | SLS | 70 | 116 |
| LC-3694-8/19 | Little Conch | offshore | 2019-8 | black(f) | BLS | AB | 70 | 117 |
| LC-3921-8/19 | Little Conch | offshore | 2019-8 | black(i) | BLR | HTA | 70 | 118 |
| LC-3989-8/19 | Little Conch | offshore | 2019-8 | black(g) | INT | SLS | 70 | 119 |
| **Batch D** | | | | | | | | |
| Normalizer-D | mix of all | mix of both | mix of all | mix of all | NA | NA | 58 | 113 |
| CR-3165-8/19 | Crocker Reef | offshore | 2019-8 | black(r) | BLR | HTA | 70 | 114 |
| CR-3679-8/19 | Crocker Reef | offshore | 2019-8 | not genotyped | INT | SLS | 70 | 115 |
| CR-3986-8/19 | Crocker Reef | offshore | 2019-8 | black(aa) | BLS | AB | 70 | 116 |
| Rocks-3105-8/19 | The Rocks | inshore | 2019-8 | skyblue | BLS | AB | 70 | 117 |
| Rocks-3148-8/19 | The Rocks | inshore | 2019-8 | skyblue | INT | SLS | 70 | 118 |
| Rocks-3906-8/19 | The Rocks | inshore | 2019-8 | skyblue | BLR | HTA | 70 | 119 |
| **Batch E** | | | | | | | | |
| Normalizer-E | mix of all | mix of both | mix of all | mix of all | NA | NA | 58 | 113 |
| Cheeca-3132-12/19 | Cheeca Rocks | inshore | 2019-12 | black(a) | BLR | HC | 70 | 114 |
| Cheeca-3916-12/19 | Cheeca Rocks | inshore | 2019-12 | lightgreen | BLS | HC | 70 | 115 |
| Cheeca-3939-12/19 | Cheeca Rocks | inshore | 2019-12 | darkred | INT | HC | 70 | 116 |
| Rocks-3105-12/19 | The Rocks | inshore | 2019-12 | skyblue | BLS | HC | 70 | 121 * |
| Rocks-3148-12/19 | The Rocks | inshore | 2019-12 | skyblue | INT | HC | 70 | 118 |
| Rocks-3906-12/19 | The Rocks | inshore | 2019-12 | skyblue | BLR | HC | 70 | 119 |
| **Batch F** | | | | | | | | |
| Normalizer-F | mix of all | mix of both | mix of all | mix of all | NA | | 58 | 113 |
| CR-3165-12/19 | Crocker Reef | offshore | 2019-12 | blackI | BLR | HC | 70 | 114 |
| CR-3679-12/19 | Crocker Reef | offshore | 2019-12 | not genotyped | INT | HC | 70 | 115 |
| CR-3986-12/19 | Crocker Reef | offshore | 2019-12 | black(aa) | BLS | SCTLD | 70 | 116 |
| LC-3694-12/19 | Little Conch | offshore | 2019-12 | black(f) | BLS | SLS | 70 | 117 |
| LC-3921-12/19 | Little Conch | offshore | 2019-12 | black(i) | BLR | HC | 70 | 118 |
| LC-3989-12/19 | Little Conch | offshore | 2019-12 | black(g) | INT | HC | 70 | 119 |

\* Used in place of label 117.

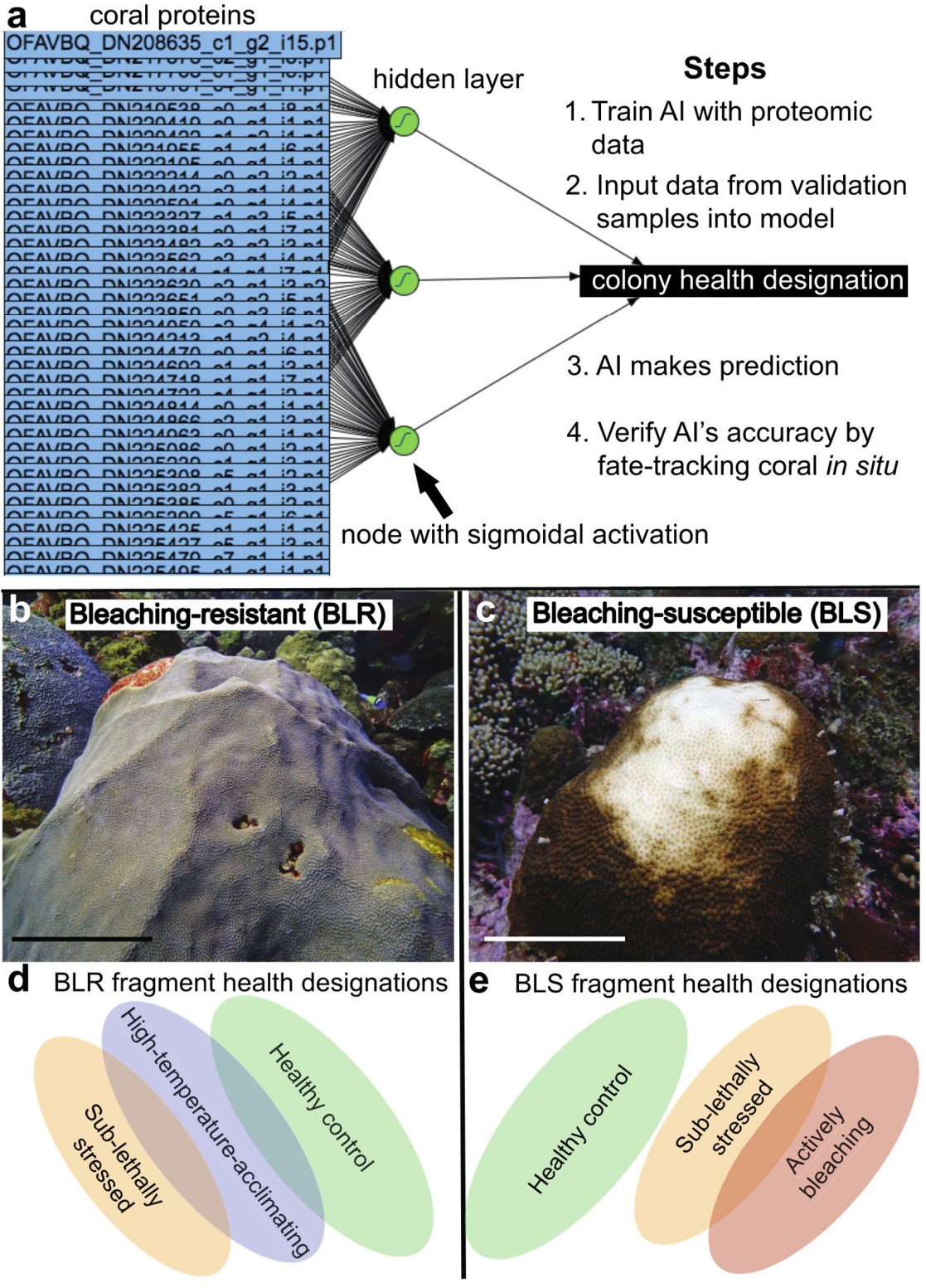

**Figure 2.** A schematic of the approach and an explanation of key terminology. Note that the neural network of panel a represents a simplified version of model "CHD-lab➔CHD-lab (**a**)" of Table 3. The full accession number is masked for all but the first protein. Representative bleaching-resistant (BLR; (**b**)) and bleaching-susceptible (BLS; (**c**) corals imaged in August 2019 have been shown (scales bars represent 30 & 20 cm, respectively). The possible fragment health designations for each colony health designation have been shown as ellipses (**d**,**e**), respectively), whose degree of overlap is based on hypothetical physiological similarity rather than actual sample sizes (see Tables 1 and 2).

### 2.3. Field Coral Fate-Tracking and Sampling

To assess the efficacy of the models to predict colony bleaching susceptibility in situ, tagged and genotyped *O. faveolata* colonies located at two inshore (Cheeca Rocks [n = 10 tagged & genotyped colonies] and The Rocks [n = 6]) and two offshore (Little Conch [n = 10] and Crocker Reef [n = 5–6]) sites were photographed alongside a Coral Reef Watch (CRW) reference card [14] and sampled on July 10th (pre-bleaching), August 21st (peak bleaching period), October 9th (post-bleaching recovery period), and December 4th of 2019. Note that biopsies from Crocker Reef represent true "test" samples for the CHD-lab➜CHD-field models since no data from corals of this site were incorporated in initial model-building. When image quality was poor and the CRW color score data could not be interpreted with confidence, dinoflagellate endosymbiont densities were instead estimated from either host/symbiont DNA (qPCR-derived [15]) or protein ratios. Images of all sampled colonies at each of the four sampling times have been archived at NOAA's National Centers for Environmental Information (accession: 0243645). Colonies were sampled with a hammer and chisel (~50 mg biopsies, or 10–20 polyps) and immersed in a liquid nitrogen dry shipper (−150 °C) prior to transport to the lab. Proteins were extracted from a subset of 120 of the 128 biopsies, and a subset of 36 samples from July, August, and December was analyzed using iTRAQ/nano-LC/MS (both described below).

### 2.4. Proteome Profiling

Proteins were extracted from the 36 field coral samples in an identical manner as for the lab samples ([11] and Supplementary File S1). There were several differences, however, between the lab and field iTRAQ protocols. First, only seven samples were analyzed in a batch for the field samples; the final label, 121, was not used except in one instance resulting from the manufacturer sending an empty tube of label 117 (Table 2). Secondly, 58 and 70 µg of protein were analyzed for the "batch normalizers" and coral samples, respectively. The former was made by mixing 6.5 µL (1.9 µg/µL) from each of the 36 samples and served to control for batch-to-batch variation. Finally, unlike for the lab samples [11], proteins were *not* randomized across batches (A–F in Table 2); this is because, even when using a batch-normalizing control sample (label 113), there was still extensive variation among batches (Supplementary File S2). Therefore, one representative sample from each colony health designation was analyzed for each of two sites (one inshore & one offshore) for one of the three sampling dates (July [pre-bleaching], August [during-bleaching], or December [post-bleaching] 2019) in a batch (Table 2). Samples from July, August, and December were therefore analyzed in batches A–B, C–D, and E–F, respectively. Note that the October "recovery" samples/data are not discussed herein. Details on iTRAQ, mass spectrometry, and analysis of the associated data can be found in Supplementary File S1.

### 2.5. Proteomic Predictive Modeling

Several descriptive analyses were taken to look at seasonal variation in coral proteomes to determine whether there was indeed sufficient variation in proteome biology across the year (& in response to the 2019 bleaching event) to even warrant predictive modeling. First, principal components analysis (PCA) and multi-dimensional scaling (MDS) were used to depict relationships and similarity among samples, respectively, and the first 3–4 coordinates from the latter analysis (depending on whether only host coral, only Symbiodiniaceae, or all holobiont proteins were included; see Table S1.) were used in a non-parametric MANOVA to document sources of proteomic variation. PLS was also used with the field coral proteins as Y's and the various field parameters as X's (including site, date, coral genotype, & coral phenotype). All statistical analyses were performed with JMP® Pro 16 or 17, with links to predictive model scripts found within Supplementary File S2 (& hosted on the open-access repository JMP Public).

JMP Pro's "model screening" platform was used to test the following modeling types with colony health designation as Y: bootstrap forest, DA, generalized regression (gen-reg), k-nearest neighbors, naïve Bayes, neural networks, PLS, stepwise regression, support vector

machines, and XGBoost. When the best model (ranking schematic described below) was a neural network, an automated GUI (as a JMP add-in) developed by D. Schmidt was used to build an additional 1000–10,000 models in which different levels of the following model input parameters were tested in randomized fashion: number of hidden layers (1 or 2), type of activation (sigmoidal, linear, or radial), number of activation nodes for each activation type (1, 2, 3, or 4), number of boosts (single-hidden layer models only; 0–20), learning rate (only when using boosts; 0–0.3), covariate transformation (yes vs. no), robust fit (used or unused), and number of tours (1–100). The penalty method was always set to "weight decay." Note that because multiple tours were used, it was necessary to perform simulations (typically 10–20) to ensure that neural network accuracies were stable; mean accuracies across all simulations have been presented in the manuscript's tables.

### 2.5.1. Lab-Trained Models

The first CHD-lab➜CHD-field model ("a;" Table 3) was trained and validated with the lab coral samples (n = 14 & 5, respectively), then tested with the field coral colonies sampled before a bleaching event in 2019 (July; n = 12). Note that the total number of lab corals used in training (n = 19) is lower than shown in Table 1 because one sample (B5-7) was compromised during processing and the bleaching resilience of another (D6-6) could not be determined due to a lack of experimental ramets from the same genet. Because 86 and 16 lab and field coral proteins, respectively, passed QC, and only 5 of these were common to both datasets due to the stochastic nature of proteomics, the data from these five proteins alone were included. In "b" of Table 3, all 19 lab samples were used for training, with 24 July (pre-bleaching) and August (during-bleaching) samples used for validation; only four proteins are featured in this model (the aforementioned five minus one absent from the August dataset). Two additional combinations of lab training and field validation/test samples were also tested (Supplementary File S2).

### 2.5.2. Field-Trained Models

In the "CHD-field➜CHD-field" models, only field coral proteomic data featured in model training. The primary differences among the many models constructed are the samples used to train, validate, and, in certain cases, test. In the simplest model, only pre-bleaching samples (July) were used for training and validation. Models were scored first in terms of their accuracy, as validated by the responses of the field colonies in situ (Figure 2a); if the AI guessed that a colony would be bleaching-susceptible based on proteomic data from the July biopsy, and this colony bleached in the 2019 August bleaching event, then the AI's guess was deemed correct. Note that December (post-bleaching) samples were included in both training and validation models because it was critical to ensure that the AI could differentiate bleaching susceptibility *after* a prior bleaching event (in which case corals may have acclimatized or otherwise undergone fundamental changes in their cellular biology due to prior stress exposure). Accuracy was calculated across all field validation or test samples. When two models were characterized by the same accuracy, the one with the smaller difference between the training and validation accuracies was deemed superior. The complexity of the model was used to break additional ties, with the more parsimonious one "championed" given its easier interpretation and computation. Only models with accuracies >80% were considered to be potentially useful for field diagnostics and are explored in detail herein.

## 3. Results and Discussion

### 3.1. Overview of the Field Coral Proteomic Dataset

Sixteen proteins were quantified in all 36 field samples; for models in which only the July and August data were incorporated, 28 proteins could instead be used for model-building since the degree of overlap was higher. When viewing a PLS-derived correlation loading plot of data from the 16 common proteins (Figure 3), there is some degree of partitioning among corals of the three colony health designations. However, one bleaching-

susceptible colony fell within the bleaching-resistant data space, and the first two PLS factors encompassed <50% of the variation; this approach is useful for exploring patterns, then, but *not* for making predictions. There was a significant effect of the fragment health designation on the host coral protein profile (Table S1 and Figure 4c); this was driven in part by the 50% higher concentration of protein OFAVBQ_DN223604_c1_g1_i2 in healthy control corals vs. all other phenotypes. The sequenced peptide's identity could not be resolved bioinformatically, though it features multiple BRCA2 repeats. Given that it contributed >40% of the variation across fragment health designation and was down-regulated in all but healthy corals, its identity should be uncovered in the near future.

**Table 3.** Neural networks trained exclusively with lab data with accuracies > 80%. The model prediction goal was either lab colony health designation (CHD; i.e., "CHD-lab") or field coral CHD. A subset of all 86 proteins was used in all but the first model because different proteins were sequenced in the lab and field samples (5 proteins found in all lab & pre-bleaching corals sampled in situ [4 when factoring in additional sampling times]). See Supplementary File S2 for models with accuracies < 80%. Sample sizes in the bottom row represent number of simulations (necessary for neural networks featuring multiple tours).

| Model Abbreviation | CHD-Lab➔ CHD-Lab [11] | CHD-Lab➔CHD-Field(a) | CHD-Lab➔CHD-Field(c) |
|---|---|---|---|
| Model prediction goal | CHD-lab | CHD-field | CHD-field |
| Training sample size (lab corals) | 14 | 19 | 19 |
| Validation data type (#samples) | Lab corals (5) | Field corals (12) | Field corals (24) |
| Validation data months | Not applicable | Jul. | Jul. and Aug. |
| Training proteins (#) | 86 | 5 | 4 |
| Model details | TanH(3)-Boost(7) [a,b] | TanH(1)-Linear(1)-Gaussian(3) [c] | TanH(2)-Linear(2)-Gaussian(4) |
| Accuracy (%) ± std. dev. | 100±1 (n = 20) | 86 ± 10 (n = 40) | 82 ± 7 (n = 40) |

[a] A simpler generalized regression model (ridge) yielded comparable accuracy. [b] See Figure 4b for the unboosted TanH(3) diagram. [c] See Figure 4d.

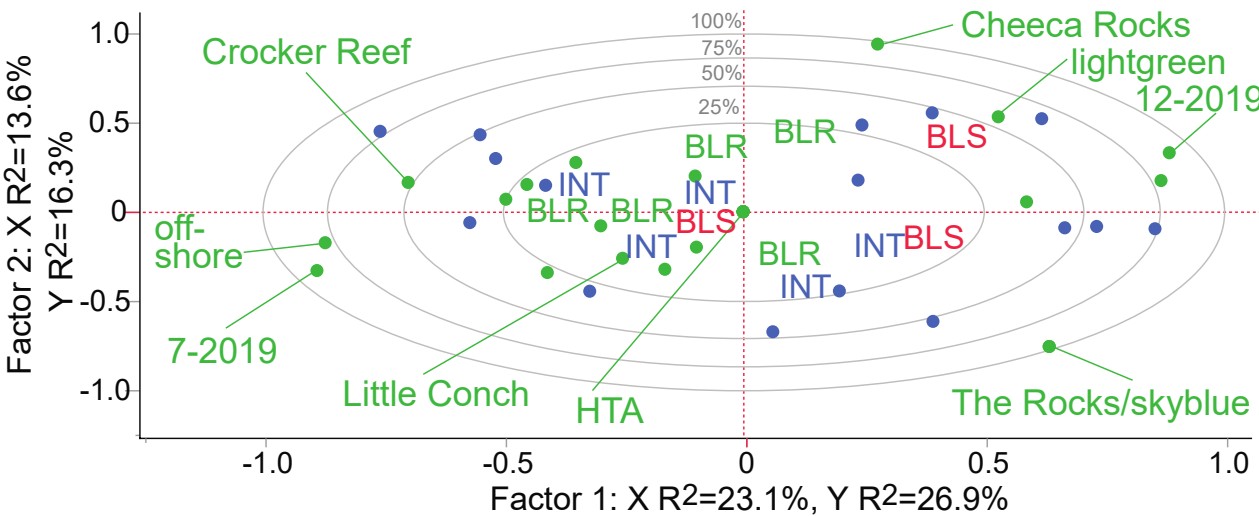

**Figure 3.** Partial least squares correlation loading plot. The 16 proteins measured in all field samples were the model Y's (blue dots) while the following were considered as putative drivers of variation (green dots, of which several have been labeled): reef site (Cheeca Rocks, The Rocks, Little Conch, & Crocker Reef), shelf (inshore vs. offshore), date (Jul. & Dec. only), host genotype (black(a), black(f), black(g), black(i), black(r), skyblue, darkred, lightgreen, or unknown), fragment health designation

(healthy control, high-temperature-acclimating [HTA], sub-lethally stressed, & actively bleaching), and colony health designation (CHD; bleaching-susceptible [BLS], bleaching-resistant [BLR], & intermediate [INT]). The samples themselves are labeled by their CHD (red, blue, & green for BLS, INT, & BLR, respectively) and include only 13 biopsies since only training data are shown.

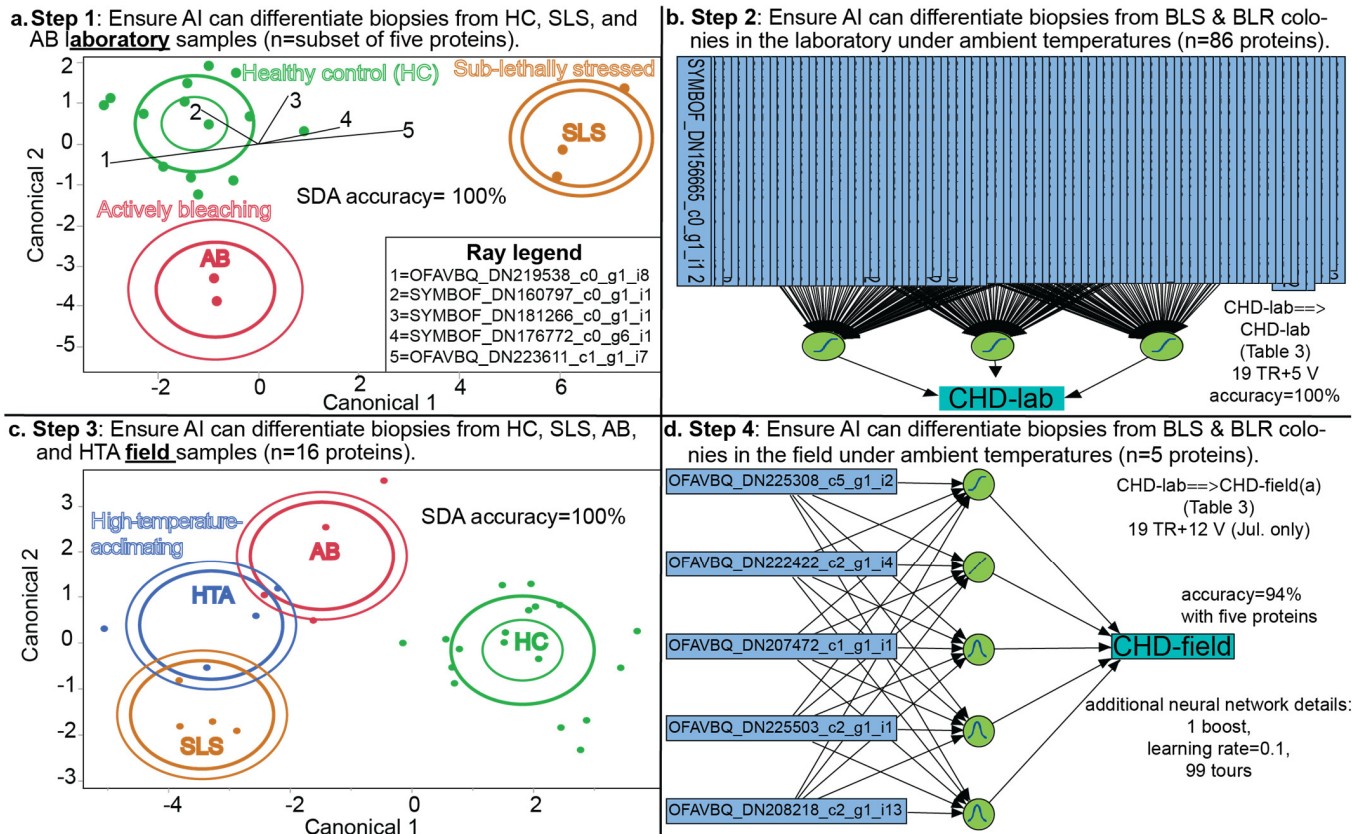

**Figure 4.** Progressive training of the AI with coral proteomic data. In the stepwise, discriminant analysis (SDA) canonical correlation plots, inner and outer ellipses represent 50 and 95% confidence, respectively, and for the latter, rays are shown only in Supplementary File S2. Note that for step 4, we have shown only one example of the "lab→field" approach in which lab-derived protein profiles were used to predict field coral bleaching susceptibility ("CHD-field"); models trained with field data exclusively are instead shown in Table 5. Note that in step 4, a third, intermediately bleaching-susceptible phenotype was considered as a model Y (vs. only bleaching-susceptible [BLS] vs. bleaching-resistant [BLR] as possible colony health designations [CHD] in the lab samples). AB = actively bleaching, HC = healthy control, HTA = high-temperature-acclimating, SLS = sub-lethally stressed.

### 3.2. Coral Health Predictions Overview

The primary goal of this work was to use protein signatures to predict whether a coral would later bleach under high temperatures. First, we used lab data exclusively to ensure that the AI could resolve differences in fragment health designations, and the corresponding PLS-based stepwise DA (Figure 4a) with a subset of only 5 of the 86 lab coral proteins that passed all QC [11] could differentiate coral biopsies of the three phenotypes with 100% accuracy. Even when withholding data from the AI, this simple model could accurately call the fragment health designation of laboratory tissue biopsies based on their protein signatures alone.

However, differentiating a bleaching coral biopsy from a healthy one is computationally simple vs. building a robust predictive model for a property of a colony or genotype in which disparate phenotypes must be considered. The colony health designation predictions included all possible fragment health designations for each queried colony (Figure 2d,e);

the AI then had to search for patterns through the "noise" associated with the significant differences in phenotypes that emerged due to temperature stress in the instance of the intermediate and bleaching-susceptible corals. This was less of an issue with the bleaching-resistant corals since, theoretically, their protein signatures would be more consistent across temperatures (Figure 2d). This is one reason why DA, MANOVA, and other descriptive approaches excel in *differentiating* fragment health designations, but not in *predicting* intrinsic properties of the colonies/genotypes from which these biopsies were taken. To resolve such complexity in a way that is useful for predicting the behavior of coral genotypes about which nothing is known, more advanced predictive modeling approaches are warranted. In the lab samples only, a neural network (Figure 4b and Table 3) could resolve colony health designations with 100% accuracy; regardless of the fragment health designation of the biopsy, the AI could accurately determine whether it was from a bleaching-resistant or bleaching-susceptible genotype (no intermediately bleaching-susceptible genotypes were identified in the lab study).

### 3.3. Coral Health Predictions from Lab Data

Prior to using the lab-derived models to predict field coral bleaching susceptibility, we first ensured that the AI could distinguish field corals of the various fragment health designations (Figure 4c); note that this analysis featured the fourth, high-temperature-acclimating fragment health designation. As with the lab data, PLS-DA could resolve these phenotypes at 100% accuracy when using only 16 proteins. In contrast, PLS-DA would *not* be effective at predicting the colony health designation (Figure 3); indeed, the *p*-value derived from the NP-MANOVA of the 16 proteins versus the three field coral colony health designations was 0.94 (Table S1). In contrast, a neural network (Figure 4d) trained with only those five host coral proteins in both the lab and field coral datasets was 94% accurate. This finding means that, were one to measure the concentrations of these five proteins in a random *O. faveolata* sample that had not yet manifested any signs of bleaching and input these data into the neural network depicted in Figure 4d (& described in Table 3 as "CHD-lab➔CHD-field(a)"), the AI would correctly predict whether the coral would bleach (bleaching-susceptible), partially bleach (intermediate), or resist bleaching in 94% of instances.

We therefore took a more detailed look at this single-hidden layer model and ran 20 additional simulations in which both training and validation data were used to calculate the misclassification rate (representing a more conservative approach). Upon doing so and then averaging the collective misclassification rates across all 40 simulations (including the original 20, in which misclassification rates were only calculated from the validation samples held back from the AI), the resulting accuracy was 86% (Table 3); although having decreased slightly, this predictive confidence nevertheless surpassed our 80% threshold.

A dependent resampled inputs analysis was undertaken to determine which of the five proteins was more heavily weighted in the model, and a DELTA-actitoxin involved in prey envenomation was characterized by a two-fold greater effect size than the second most influential protein (an E3 ubiquitin protein ligase; Table 4). Although it is tempting to develop a story of coral resilience around prey caption and heterotrophy based on this finding, it is important to note that, in a non-parametric one-way ANOVA, the concentration of this protein was statistically significant only at an alpha of 0.05 (*p* = 0.03; Table 4), and the difference between bleaching-resistant versus bleaching-susceptible corals was only 50% (former > latter). In other words, concentrations of this protein in isolation would be insufficient to resolve differences in bleaching susceptibility, even if it does imply a role of coral heterotrophy (which could be hypothesized to benefit corals that have been photosynthetically compromised on account of prolonged high-temperature exposure [16]).



**Table 4.** Subset of host coral (OFAV) and Symbiodiniaceae (SYMBOF) proteins featured in proteomic predictive modeling analyses. All *e*-values were $<10^{-80}$, and the top hit was always a scleractinian coral for the host (typically *O. faveolata* or *Acropora millepora*). The "model total effect" was derived from a dependent resampled inputs analysis. AB = actively bleaching. CHD = colony health designation. HC = healthy controls. NA = not applicable.

| Accession | Protein Name | Top BLAST Hit Accession | Protein Function | CHD-Lab→CHD-Field(a) Model Total Effect |
|---|---|---|---|---|
| **Proteins featured in model CHD-lab→CHD-field(a) (Table 3)** | | | | |
| OFAVBQ_DN222422_c2_g1_i4 [a] | DELTA-actitoxin | **XP_044182609.1** | prey capture | 0.44 |
| OFAVBQ_DN225308_c5_g1_i2 | E3 ubiquitin protein ligase | **XP_020616505.1** | protein degradation | 0.25 |
| OFAVBQ_DN208218_c2_g1_i13 | histone-lysine N-methyltransferase SETD1B | **XP_020616327.1** | transcription | 0.20 |
| OFAVBQ_DN225503_c2_g1_i1 [b] | concanavalin A-like lectin/glucanase | **XP_020611325.1** | cell binding/immunity | 0.19 |
| OFAVBQ_DN207472_c1_g1_i1 | histone H2A | **XP_020604133.1** | chromatin | 0.18 |
| **Other proteins of interest** | | | | **Finding** |
| OFAVBQ_DN223604_c1_g1_i2 [c] | unknown | **XP_020618813.1** | unknown | 50% lower in Aug. |
| SYMBOF_DN75265_c0_g1_i2 | chromosome segregation protein SMC | **CAI3980295.1** | cell division | see Table 5. |
| OFAVBQ_DN221258_c1_g2_i8 | F-actin-methionine sulfoxide oxidase | **XP_020606640.1** | molecular trafficking | see Table 5. |
| OFAVBQ_DN220777_c1_g3_i5 | titin-like | **XP_020600883.1** | various [d] | see Table 5. |
| OFAVBQ_DN220189_c1_g1_i3 | unknown | **XP_020619646.1** | unknown | see Table 5. |
| OFAVBQ_DN186152_c0_g1_i1 | transcription factor AP1 | **XP_020624642.1** | transcription | see Table 5. |

[a] BLR > BLS (~50%; *p* = 0.03). [b] AB > HC (1.5-fold) in lab samples [11]. [c] HC > all other fragment health designations. [d] Involved in musculature in vertebrates, though thought to play a role in chromatin in invertebrates.

### 3.4. Coral Health Predictions from Field Data

An exclusively field-based analysis was next undertaken in which replicate colonies from diverse genotypes at sites of varying oceanographies were sampled before, during, and after a bleaching event. The same BRCA2-rich protein discussed above that contributed significantly to variation across coral biopsy fragment health designations also contributed 36% to the overall temporal variation (see Supplementary File S2 for other univariate trends in situ). Given that its cellular concentrations in August were 50% lower (Table 4), this signifies that this currently unidentifiable protein must be prioritized by those interested in the cellular biology of this important Caribbean reef-builder.

One could argue that it is only practical to validate the model with data from corals that have not yet bleached since this is how it would be used by a researcher or conservationist in the field; there would be no conservation merit in inputting data from an actively bleaching coral into the model since the bleaching susceptibility of that coral would inherently be known. The model's value would be in leveraging data from corals that have not yet bleached, particularly those that may be sub-lethally stressed but not yet manifesting visible signs of bleaching, and then determining whether these corals later bleached. Models trained with field data from pre-bleaching samples only ("July data only(b)" in Supplementary File S2) were characterized by accuracies of 92–93% with neural networks and support vector machines. Although this signifies that the AI could differentiate a bleaching-prone coral from a bleaching-susceptible one *before* the former manifested any signs of bleaching, we were concerned about the timing since the models were validated with proteomic data from corals sampled only one month before the bleaching event began,

at which point 2 DHWs had already accrued (Figure 1); while visibly healthy, these corals may already have actually been sub-lethally stressed.

For this reason, it was necessary to build a large number of additional models in which various proportions of corals of differing colony health designations of different genotypes sampled at different reefs on different dates were featured as training, validation, and test samples (Table 5). The overall goal was to effectively "trick" the AI to reduce the chances of over-fitting, a significant concern with neural networks in particular, since the AI will repeatedly "comb" the predictors to try and find combinations (& under varying degrees, levels, & types of activation) that lead to an accurate solution. With training data alone, virtually all neural networks tested yielded models with 100% accuracy.

When holding back data from validation and/or test samples (i.e., those whose data were not featured in the first incarnation of the model), accuracies (Table 5) ranged from 85–95% (depending on how the samples were partitioned among training, validation, & test samples). In other words, if 100 coral samples were given to a technician who had no knowledge of their stress tolerance, and the technician quantified the 16–28 proteins used in the models generated herein, he/she would accurately predict the bleaching likelihood of 85–95 of the associated coral colonies. It is important to emphasize that, because colony health designation is a property of the coral colony, it was necessary to ensure that the AI could accurately predict this trait regardless of the biopsy's phenotype. Although a bleaching-resistant coral would never yield a bleached biopsy (Figure 2d), a bleaching-susceptible coral could be associated with a healthy one (Figure 2e), and a healthy biopsy from such a colony could fundamentally differ from that of a bleaching-resistant colony (despite both being "healthy"); indeed, many have shown that many such corals display "stress-hardened" proteomes [17] and transcriptomes [18] to cope with life in marginal environments. This is why the modeling types of Table 5 are amongst the most computationally intensive in existence. In at least one instance, however, ML was not required; in the model CHD-field➜CHD-field(c), in which only pre- and post-bleaching samples were used for training, validation, and testing, gen-reg with a lasso algorithm was sufficient to accurately predict bleaching susceptibility (95%). Unlike neural networks, response variable reduction is possible with gen-reg, and the model featured only two proteins: the DELTA actitoxin described above and a Symbiodiniaceae chromosome segregation protein (SMC; accession: SYMBOF_DN75265_c0_g1_i2) that has been found to be important in high-temperature adaptation of these dinoflagellates [19].

Upon programming the AI to output the conditions that would maximize the likelihood of a coral resisting bleaching based on the gen-reg-lasso model, $z$-scores for the actitoxin and SMC protein of ~2.2 and −1.3, respectively, were yielded; a coral maintaining abnormally high concentrations of the former and abnormally low concentrations of the latter would stand a 50% chance of being bleaching-resistant. Because this degree of confidence was low, we carried out a similar desirability analysis with the top-performing neural network for CHD-field➜CHD-field(e) (Table 5), and the top 10 most influential proteins from the associated dependent resampled inputs analysis are shown in Figure 5 (as raw data & not $z$-scores). In this instance, the conditions portrayed in the plot would reflect a bleaching-resistant coral at 99% confidence. The most influential protein was, again, the actitoxin, with high concentrations of this protein being associated with bleaching-resistant corals (desirable mean of ~1.7=~2-fold higher than that of an average coral). Bleaching-resistant corals would also demonstrate higher-than-average concentrations of protein OFAVBQ_DN220777_c1_g3_i5, a titin-like protein with a wide variety of hypothesized functions. In contrast, a coral with a 99% chance of being bleaching-resistant would express low levels of OFAVBQ_DN220189_c1_g1_i3 (50% less than an average coral), an uncharacterized protein with a putative DNA-binding domain (Table 4). Increasing efforts by biologists to elucidate key facets of anthozoan-dinoflagellate endosymbioses [20] means that we may soon better understand the role of these (& other) proteins in coral thermo-adaptation, as well as which of the underlying genes are under selection [21,22].

**Table 5.** Models for predicting coral colony health designation (CHD) from field coral data exclusively. Of the 65 proteins identified in the July (pre-bleaching) samples, only 28 were also quantified in the August (bleaching event) samples, of which 16 (11 host & 5 Symbiodiniaceae) were also quantified in the December (post-bleaching) samples; 5 of these were also found in the lab samples. Models with accuracies <80% are found in Supplementary File S2. When the superior model was a neural network (NN), 10–20 simulations were run to accommodate having considered up to 100 tours, with the mean ± standard deviation shown in the cell. The "most important protein(s)" was/were derived from a dependent resampled inputs analysis. Gen-reg = generalized regression. SVM = support vector machines.

| Model Input Type | Field Corals Sampled across Bleaching Event in 2019 (Figure 1) | | | | |
|---|---|---|---|---|---|
| **Model Name** | **CHD-Field→ CHD-Field(a)** | **CHD-Field→ CHD-Field(b)** | **CHD-Field→CHD-Field(c)** | **CHD-Field→CHD-Field(e)** | **CHD-Field→CHD-Field(f)** |
| Training # | 19 | 18 | 11 | 12 | 15 |
| Training data months | Jul., Aug., Dec. | Jul. and Dec. | Jul. and Dec. | Jul. | Jul. and Aug. |
| Validation # | 8 | 12 | 4 | 12 | 6 |
| Validation data months | Jul., Aug., Dec. | Aug. | Jul. and Dec. | Aug. | Jul. and Aug. |
| Test sample # | 3 | NA | 3 | Not applicable (NA) | 3 |
| Test data months | Jul., Aug., Dec. | NA | Jul. and Dec. | NA | Jul. and Aug. |
| Training proteins | 16 | 16 | 16 | 28 | 28 |
| Model type #1 (accuracy [%]) | SVM: 90%[a] | Naïve Bayes: 94% | Gen-reg lasso: 95% | NN: 95 ± 6% (n = 20) | SVM: 94% |
| Model type #2 (accuracy [%]) | NN: 84 ± 9% (n = 40) [a] | NN: 87 ± 13% (n = 40) [a] | NN: 87 ± 14% (n = 20) | Naïve Bayes: 92% | NA |
| Most important protein(s) | OFAVBQ_DN221258_ c1_g2_i8 | OFAVBQ_DN222422_ c2 _g1_i4 | OFAVBQ_DN222422_c2_g1_i4 SYM-BOF_DN75265_c0_g1_i2 | OFAVBQ_DN186152_ c0_g1_i1 [b] OFAVBQ_DN220777_c1_g3_i5 | OFAVBQ_DN222422_c2_g1_i4 |

[a] Included all 30 samples in calculation of misclassification rate. [b] Excluded from Figure 5 since it was not featured in every simulation.

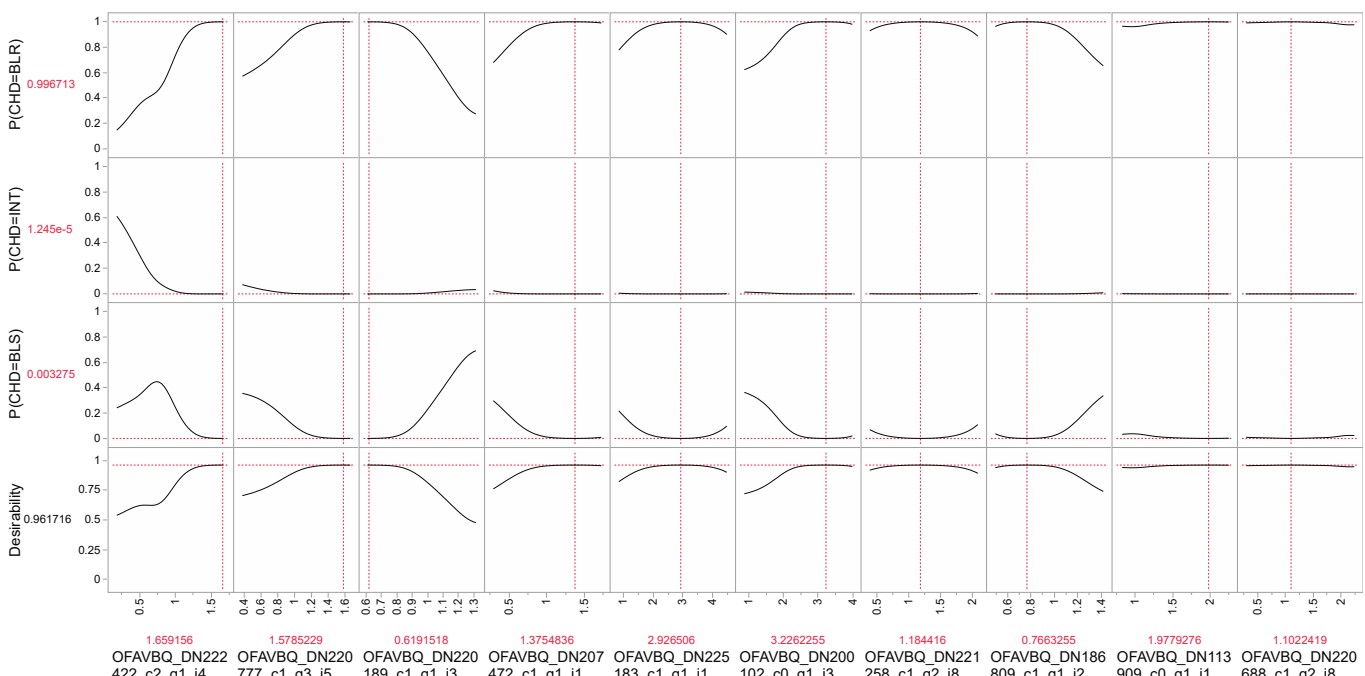

**Figure 5.** Desirability analysis based on the neural network "CHD-field➜CHD-field(e)" (Table 5). The "P" before the terms on the y-axes refers to the probability, and the protein concentration levels associated with a 99.7% chance of a coral being bleaching-resistant (BLR) are shown. The desirability approached the target value of 1 since there was <1% chance of a coral demonstrating these protein concentrations either being intermediately bleaching-susceptible (INT) or fully bleaching-susceptible (BLS). Only the top 10 most influential proteins of all 28 used in the model have been shown (ordered from most influential on the left to least on the right); values in red above protein accession numbers represent the target levels (raw iTRAQ values) that resulted in maximum desirability.

### 3.5. Other Models

We also built a large number of models in which lab and field data were combined in various ways to yield AI-driven predictions of coral bleaching susceptibility; a subset is shown in Table S2 with the complete list found in Supplementary File S2. In general, neural networks and support vector machines were required to achieve validation or test sample accuracies of 85–90%, though in some instances simpler ML models, such as the bootstrap forest, were instead capable of doing so. As for gen-reg mentioned above, bootstrap forest models are easy to interpret and permit response variable reduction, though they are often only useful for hindcasting [23] and have lower power to correctly classify true field-test samples in our experience.

Given that the lab- and field-sample-derived models were roughly similar in terms of accuracy (85–95%), which should be used? The latter are likely more robust for at least three reasons. For one, certain genotypes displayed differing colony health designations, depending on whether they were in laboratory aquaria or in situ (Table 1). This was unexpected, signifies that there is an effect of genotype on aquarium acclimation capacity, and lends support to the field-specific models since, in virtually all cases, one will be interested in predicting field (& not lab) coral behavior. Similarly, no corals of the lab study demonstrated the intermediate level of bleaching susceptibility documented in some field corals, meaning the models were fundamentally biased against this phenotype. Thirdly, when using lab data to make field coral bleaching predictions, far fewer proteins were used (4–5 vs. 16–28 with field-derived models); this will actually be an issue with this approach moving forward because, even if only 4–5 proteins are ultimately needed to make a robust prediction, there is no guarantee that they will be sequenced in each sample. An antibody-based ELISA or Western blot assay may instead be warranted.

*3.6. Caveats and Conclusions*

Although predicting the bleaching susceptibility of massive corals along a well-studied portion of the Florida Reef tract with >90% accuracy is an achievement, it is important to note several caveats. First, this analysis was undertaken with a single species, which is among the best studied in the Caribbean. Secondly, only ~10–15 genotypes were tested in a relatively small area; the accuracy is likely to taper off farther from the test sites. Aside from the stochastic nature of proteomics, this undertaking was expensive (USD 50,000 in protein sequencing costs, including reagents & instrument time) and time-consuming (four years); it would be an impractical way forward for all but the best-funded marine labs, and even then, many corals will have perished by the time similar such predictive models are constructed for a large number of keystone species [24].

We are now taking these molecular biotechnological data and "working backwards," trying to find cheaper, easier-to-measure proxies of reef resilience and coral longevity that do *not* require training in molecular biology, computer science, and invertebrate physiology [4,25,26]. The molecular biotechnology–AI approach applied herein may still have value in small areas, such as resorts with house reefs or coral nurseries, with significant financial backing and the need for high-resolution diagnostics, and GUIs derived from the models are now being placed on websites such as coralreefdiagnostics.com such that interested individuals can input data from their corals (or reefs) of interest, and the AI will make predictions of coral bleaching susceptibility. These organismal-scale predictions could then be compared to simpler, large-scale, temperature-derived predictions that constitute the basis of all current models of reef persistence [27–30] to determine the degree of congruency (if any). It is likely that the hybrid approach, in which environmental, ecological, and physiological data are integrated into a holistic model (*sensu* [25,26]), will be associated with higher accuracy for forecasting coral climate resilience than our proteomics-exclusive approach or the temperature+coral abundance-focused models featured in prior works.

**Supplementary Materials:** The following supporting information can be downloaded at: https://www.mdpi.com/article/10.3390/app13031718/s1: A supplemental file (Supplementary File S1) containing supplemental methods, supplemental results, and supplemental tables, as well as a tab-delineated data file featuring all data (Supplementary File S2). Refs. [31–33] are cited in supplementary materials.

**Author Contributions:** A.B.M. carried out all components of the analysis. C.L. provided funding support to cover the article processing charges. All authors have read and agreed to the published version of the manuscript.

**Funding:** This work was funded by NOAA (USA) through the 'Omics Initiative' (NRDD18978). Article processing charges were covered by a grant from Taiwan's Ministry of Science and Technology (MOST 110-2313-B-291-001-MY3 to C.L.).

**Institutional Review Board Statement:** A coral collection permit was issued to A.B.M. by the Florida Keys National Marine Sanctuary (USA): #FKNMS-2019-067.

**Informed Consent Statement:** Not applicable.

**Data Availability Statement:** In addition to the online supplemental data (Excel) file (Supplementary File S2), the proteomic data (as MZML & MZID files) and images of all sampled coral colonies were deposited in the University of California, San Diego's (USA) "MassIVE" data repository (accession: MSV000089240) and NOAA's National Centers for Environmental Information (accession: 0243645), respectively.

**Acknowledgments:** Thanks are given to G. Kolodziej for assistance in sample collection.

**Conflicts of Interest:** The authors declare no conflict of interest. The funders had no role in the design of the study; in the collection, analyses, or interpretation of data; in the writing of the manuscript, or in the decision to publish the results.

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
