# Peer review of "Field-Testing a Proteomics-Derived Machine-Learning Model for Predicting Coral Bleaching Susceptibility"

_applsci, doi:10.3390/app13031718_

Round 1
Reviewer 1 Report
The article is written in good language on a relevant topic. I think, this manuscript can be published in the Applied Sciences after minor revision.
1. Self-citation of the author Anderson B. Mayfield reaches 56% (articles 4,7,8,9,10,13,14,15,18,20-31,36,37). It is necessary to reduce the level of self-citation to 10-15%.
2. Why was Orbicella faveolata investigated for research?
3. The name “O. Faveolata” I recommend writing in italics throughout the text of the article.
4. The text mentions Acropora millepora, and the whole article is devoted to Orbicella faveolata. Please explain whether Acropora millepora has been studied additionally?
5. The grant number is not indicated in the funding line.
Author Response
Thank you for reviewing our article. We have provided responses to your comments in the attached Word document and have uploaded the revised version of the manuscript, as well.
Reviewer 2 Report
This is a very interesting study using AI to predict bleaching susceptibility in corals- Orbicella faveolata. The authors have presented their intensive methodology and findings and have even discussed the limitations of this model.
Please rephrase line no 71 in methodology for better clarity.
Author Response
Thank you for reviewing our article and for seeing its value. We have addressed your concern in the attached Word document by specifically attempting to rephrase the line in question.
Reviewer 3 Report
The main goal of this paper is to use the protein characteristics to predict whether coral will bleach under high temperature. This work uses the special data of the laboratory. The author uses molecular biotechnology+AI approach and obtains exciting experimental results.
This article has clear logic and detailed experiment. I believe that the research result will be widely applied in practice. I think this article should be accepted.
Minor modification problem:
It is suggested to draw an experimental flow chart to express the experimental process more clearly.
Author Response
This is actually an excellent suggestion, and we had thought to include a conceptual figure before. I have now made a new figure, Figure 2, that shows the conceptual approach, some representative images, as well as the visual distinction between the colony health designation and the fragment health designation, which is critical for understanding the article. Hopefully, with this new figure, it is more obvious what we set out to accomplish.